# Ovarian Causes of Pseudomyxoma Peritonei (PMP)—A Literature Review

**DOI:** 10.3390/cancers16081446

**Published:** 2024-04-09

**Authors:** Sinziana Ionescu, Marian Marincas, Octavia Luciana Madge, Irinel Gabriel Dicu-Andreescu, Elena Chitoran, Vlad Rotaru, Ciprian Cirimbei, Mirela Gherghe, Adina Ene, Robert Rosca, Madalina Radu, Laurentiu Simion

**Affiliations:** 1Surgery Department, “Carol Davila” University of Medicine and Pharmacy, 050474 Bucharest, Romania; ionescu_sinzy@yahoo.com (S.I.); elenachitoranphd@gmail.com (E.C.); vladrotaruphd@gmail.com (V.R.); cipriancirimbeiar@gmail.com (C.C.); dr.simion.laurentiu@gmail.com (L.S.); 2General Surgery and Surgical Oncology Department I, Bucharest Institute of Oncology “Prof. Dr. Al. Trestioreanu”, 022328 Bucharest, Romania; octaviamadge@mail-box.org (O.L.M.); igdandreescu@gmail.com (I.G.D.-A.); 3Faculty of Letters, University of Bucharest, 030018 Bucharest, Romania; 4The Clinical Nuclear Medicine Laboratory, Oncological Institute “Prof. Dr. Alexandru Trestioreanu”, 022328 Bucharest, Romania; 5Pathology Department, Oncological Institute “Prof. Dr. Alexandru Trestioreanu”, 022328 Bucharest, Romania; adinaenedr@gmail.com (A.E.); madalinar405@gmail.com (M.R.); 6Pathology Department, Bucharest Emergency University Hospital, 050098 Bucharest, Romania; dr.robertrosca@proton.me

**Keywords:** pseudomyxoma peritonei, ovarian cancer, gelatinous disease, cancer, oncology, surgical oncology, mucinous disease, surgery, gynecology, peritoneal metastasis

## Abstract

**Simple Summary:**

Pseudomyxoma peritonei (PMP) is a rare, slow-growing, and poorly understood neoplasm. It is characterized by varying degrees of malignancy and the production of mucinous, gelatinous structures. The development of PMP is typically associated with the rupture of appendiceal mucinous tumors and other gastrointestinal or ovarian mucinous tumors. This present literature review was conducted to better describe the ovarian causes of PMP. The main instances in which PMP can have an ovarian cause include the following: mucinous cystadenoma, mucinous ovarian cancer, colon cancer with ovarian metastasis, malignant transformation of an ovarian primary mature cystic teratoma, appendiceal mucocele with peritoneal dissemination, mucinous borderline tumor developing inside an ovarian teratoma, and the association between a mucinous bilateral ovarian cancer and a colonic tumor. We undertook a literature study to identify and underline the feasible treatments for PMP, since the limited knowledge of this cancerous condition raises the likelihood of delayed diagnosis or progressive deterioration.

**Abstract:**

Background. Pseudomyxoma peritonei (PMP) is a rare, progressive, slowly growing, inadequately understood neoplasm with a 5-year progression-free survival rate of as low as 48%. It is characterized by varying degrees of malignancy and the production of mucinous and gelatinous structures. Typically, the development of pseudomyxoma peritonei is associated with the rupture of appendiceal mucinous tumors and other gastrointestinal or ovarian mucinous tumors. The goal of our literature review was to identify various aspects that characterize the ovarian causes of pseudomyxoma peritonei. Materials and methods. The authors performed an extensive literature search between 1 February 2024 and 2 March 2024 on the following databases: Pubmed, Scopus, Oxford Journals, and Reaxys, and the findings were summarized into seven main clinical and paraclinical situations. Results. According to our research, the main instances in which pseudomyxoma peritonei can be triggered by an ovarian cause are the following: (1) mucinous cystadenoma; (2) mucinous ovarian cancer; (3) colon cancer with ovarian metastasis; (4) malignant transformation of an ovarian primary mature cystic teratoma; (5) appendiceal mucocele with peritoneal dissemination mimicking an ovarian tumor with peritoneal carcinomatosis; (6) mucinous borderline tumor developing inside an ovarian teratoma; and (7) the association between a mucinous bilateral ovarian cancer and a colonic tumor. Conclusions. In our study, we aimed to provide a comprehensive overview of the ovarian causes of pseudomyxoma peritonei, including its epidemiology, imagery characteristics, symptoms, current treatment, and promising future therapies, in the hopes of finding feasible solutions, as a lack of understanding of this mucus-secreting malignant disease increases the risk of delayed diagnosis or uncontrolled deterioration.

## 1. Introduction

Pseudomyxoma peritonei (PMP) is an uncommon abdominal illness characterized by symptoms such as abdominal discomfort [1] and mass, exhaustion, and weight loss. The condition is defined by a gradual buildup of mucin in the peritoneal cavity, along with the existence of gelatinous ascites and peritoneal implants. Because the symptoms are nonspecific, it is often identified coincidentally during surgery for other reasons (Figure 1, Figure 2 and Figure 3 show the intraoperative aspects of pseudomyxoma peritonei).

The condition is more prevalent in women than in men, with an estimated frequency of one to two cases per million per year, as reported by Tsoukalas [2]. It is associated with malignant tumors in the appendix (see Figure 4 and Figure 5); however, there is a debate over the specific epithelium from which it arises. The hypothesis that ovaries might be an alternative main location of PMP genesis is regarded as plausible, but lacks a complete explanation in terms of immunochemistry [3] (see Figure 6 and Figure 7). In a study designed by Yan [4], the tumors exhibited positivity for CK20, CDX2, CEA, and Villin. SATB2 showed broad positive expression in teratoma-associated ovarian mucinous tumors and was negative in original ovarian mucinous tumors. Varied expression of MUC was seen in various tumors. Immunohistochemistry for synaptophysin supports the histopathological differential diagnosis in some cases, as underlined by Mukherjee [5]. Some oncologists and pathologists use the term “PMP” to describe any disorder involving gelatinous material in the abdomen and pelvis.

The pattern of insidious growth of PMP, which makes the tumor asymptomatic or paucisymptomatic until the disease has reached advanced stages, makes it similar (in this clinical behavior and in its rare incidence) to a neuroendocrine tumor [6].

## 2. Materials and Methods

The authors performed an extensive literature search between 1 February 2024 and 2 March 2024 on the following databases: www.Pubmed.gov, www.scopus.com, www.oxfordjournals.com, and www.reaxys.com, and the findings were summarized into seven main clinical and paraclinical situations (as explained in Section 3). The parameters of the search performed on PubMed were: “Pseudomyxoma AND (ovarian OR ovary)”, published in the last 5 years, in English, with the research done on humans. The second literature research, performed on Scopus had the following selection criteria: subject of medicine, published between 2019 and 2024, in English, and the articles had as keywords of the search, “Pseudomyxoma”, “Pseudomyxoma peritonei”, or “Peritoneal pseudomyxoma”. A third search was performed on oxfordjournals.com, in the fields “medicine and health”, published in the last 5 years. The fourth and last literature search was done on the Reaxys database, and the words were “pseudomyxoma peritonei” AND “ovary” OR “ovarian”, published between 2019 and 2024.

## 3. Results

### 3.1. Ovarian Causes of PMP

The main instances in which pseudomyxoma peritonei can be triggered by an ovarian cause are the following: (1) mucinous cystadenoma; (2) mucinous ovarian cancer; (3) colon cancer with ovarian metastasis; (4) malignant transformation of an ovarian primary mature cystic teratoma; (5) appendiceal mucocele with peritoneal dissemination mimicking an ovarian tumor with peritoneal carcinomatosis; (6) mucinous borderline tumor developing inside an ovarian teratoma; and (7) the association between a mucinous bilateral ovarian cancer and a colonic tumor.

#### 3.1.1. Mucinous Cystadenoma

In his research, Purwoto [7] emphasized that mucinous cystadenoma could be found in 10–15% of ovarian tumors. Diagnosis and therapy should be determined accurately due to the potential progression to pseudomyxoma peritonei (PMP).

#### 3.1.2. Mucinous Ovarian Cancer

Tjokroprawiro [8] documented a case of an exceptionally large and distinctive pseudomyxoma peritonei. A 52-year-old woman with recurrent mucinous ovarian cancer stage IIIC attended hospital presenting with an abdominal tumor. The CT scan revealed the presence of mucinous fluid buildup and an incisional hernia. A surgery was conducted to remove 21 L of mucinous fluid and excise a peritoneal pocket. The patient had a successful recovery and was discharged from the hospital on the seventh day after the procedure. One month post-surgery, the patient attended the emergency hospital with symptoms of dyspnea. The chest X-ray revealed bilateral pneumonia associated with COVID-19 infection. The patient died as a result of the condition worsening.

Yeom [9] documented a case of hypercalcemia linked to a primary mucinous ovarian tumor followed by PMP, resulting in a fatal outcome.

#### 3.1.3. Colon Cancer with Ovarian Metastasis

Tumors originating from the ovary and appendix have been extensively recorded in cases of pseudomyxoma peritonei (PMP), sparking ongoing discussions on the source of the tumors. Typically, these tumors originate from a single main source, often the appendix, and then migrate to the ovaries. Sarkar’s research [10] found that in 80% of instances, ovarian masses with mucinous ascites are predominantly metastatic. A comprehensive literature study conducted by Yehya [11] indicated that patients should be indicated an appendectomy in addition to their original TAH-BSO because of two main, non-synchronous lesions. This would ensure precise staging and ascertain if the two neoplasms originated independently or were metastatic. New genetic indicators, such as the Special AT-rich sequence-binding protein 2 (SATB2) have been identified. SATB2 is present in individuals with a LAMN and absent in those with a primary mucinous ovarian cancer. The open question for surgeons would be: should we prophylactically remove the appendix in the setting of mucinous ovarian tumors? Matsuzono’s paper [12] states that appendectomy should be performed if the appendix looks unusual or if pseudomyxoma peritonei (PMP) is detected.

For peritoneal surface malignancy, the main staging tool is the peritoneal cancer index (PCI). The prognostic value of PCI varies in PMP. Chua [13] stated that while a PCI limit may be considered for more severe signet ring cell carcinoma of the appendix, there is currently no limit for all PMP cases caused by primary appendiceal cancer.

According to the indications of Pseudomyxoma Centers, we would like to emphasize the importance of taking peritoneal biopsies, performing appendicectomy, and draining the ascites, when confronted with a PMP with a likely colon origin. A further patient referral to a tertiary center for proper surgical care (mainly HIPEC + CRS) is mandatory in order to increase the chance of a good treatment response.

#### 3.1.4. Malignant Transformation of an Ovarian Primary Mature Cystic Teratoma (MCT)

Balakrishnan [14] reported a low-grade mucinous neoplasm that developed from a mature ovarian teratoma and resulted in pseudomyxoma peritonei.

Ponzini [15] described a case of a diffuse peritoneal adenomucinosis (DPAM) variant of PMP originating from a ruptured ovarian primary MCT that transformed into a low-grade appendiceal-like mucinous neoplasm.

Csanyi Bastien [16] assumed that, when PMP originates from a mucinous ovarian lesion without appendiceal involvement, it likely had a digestive teratomatous origin. This highlighted the need to aggressively look for teratomatous symptoms in cases of ovarian PMP.

#### 3.1.5. Appendiceal Mucocele with Peritoneal Dissemination Mimicking an Ovarian Tumor with Peritoneal Carcinomatosis

In a literature review, Singh [17] underlined that appendiceal mucocele in females might resemble adnexal pathology, presenting as a detectable pelvic mass. This can be difficult to identify before surgery, either during imaging or at the time of the operation. The early presentation may include nausea, vomiting, changes in bowel habits, gastrointestinal bleeding, and genitourinary symptoms. Appendiceal mucocele may lead to intestinal blockage due to intussusception.

Ayadi [18] reinforced the idea that mucocele of the appendix with pseudomyxoma peritonei is an uncommon condition that might display ambiguous signs. Having a preoperative diagnosis is crucial for guiding proper treatment and reducing the risk of problems during and after surgery.

Gallardo-Martinez [19] found that identifying the source of a pseudomyxoma peritonei is difficult since the appendix and ovaries are often afflicted at the same time. Alghamdi [20] emphasized that the appearance of appendiceal mucocele is often nonspecific due to its anatomical location. Physicians should include it in the list of possible causes for a chronic growing ovarian cyst or adnexal tumor.

Appendiceal mucinous neoplasms often have peritoneal metastases when the initial tumor in the appendix is diagnosed. However, liver metastases and lymph node metastases are rare. Sugarbaker’s [21] research on lymph-node positive pseudomyxoma peritonei concluded that long-term survival is rare but may be seen in certain individuals. Neoadjuvant chemotherapy response significantly influences the likelihood of a positive result, and the importance of identifying the lymphatic dissemination in the case of an oncological disease was emphasized also by Voinea [22].

#### 3.1.6. A Mucinous Borderline Tumor Developing inside an Ovarian Teratoma

Mucinous borderline tumors often present as bigger, multi-cystic masses that frequently affect just one side. Borderline mucinous tumors may also manifest as pseudomyxoma peritonei, posing a challenge to differentiating them from malignant mucinous carcinoma, as shown by Flicek [23].

Examinations of pseudomyxoma from the appendix revealed that KRAS and GNAS pathogenic mutations are prevalent genetic characteristics of pseudomyxoma peritonei. Identifying the origin of the tumors only via genetic mutations is challenging. Taguchi [24] conducted research on a case of pseudomyxoma peritonei of ovarian origin. The diagnosis was made using thorough genomic profiling with ploidy analysis on main, recurring, and autopsy tumor tissues. The research provided histological and genetic evidence that the main ovarian borderline tumor originated from the intestinal component of an ovarian teratoma, and that the following pseudomyxoma peritonei developed from the primary ovarian tumor. Integrative genomic analysis proved helpful for determining the cellular source of tumors and accurately understanding the disease course.

Teymoordash’s study [25] revealed that mucinous borderline tumors often recur after two years; however, in this particular instance, the relapse occurred after 5.5 years as pseudomyxoma with borderline pathology.

Ryu [26] documented an unusual instance of PMP with clinicopathological characteristics that bridge PMP and borderline mucinous tumors originating in a mature cystic teratoma of the ovary.

#### 3.1.7. The Association between a Mucinous Bilateral Ovarian Cancer and a Colonic Tumor

In research presented by Nofal [27], the prevalence of several primary malignancies in a single patient ranged from 2% to 7%. It may be challenging to determine whether the second tumor is a separate primary tumor or a metastatic growth from the first tumor, as described in this study. Primary colorectal cancer and primary ovarian tumors have been shown in several studies. The prevalence of cancer in other organs in women with ovarian cancer is reported to be 2.8%. The most common combinations are ovarian cancer with lung cancer or ovarian cancer with colon cancer.

### 3.2. General Features of PMP with an Ovarian Origin

#### 3.2.1. Mutations and Prognostics

As presented also by Rufian-Andujar [28], both the Ronnett and PSOGI classifications accurately predicted the survival for patients with PMP. When modified by the CCS (completeness of cytoreduction score), the PSOGI classification showed greater predictive availability for OS compared to the Ronnett classification. Luque-Gonzalez [29] argued that having a single reliable classification for pseudomyxoma peritonei is crucial for sharing knowledge and developing diagnostic and treatment protocols to ensure timely and effective patient care.

Pseudomyxoma peritonei (PMP) is an uncommon cancer categorized by the Peritoneal Surface Oncology Group International (PSOGI) classification. The treatment response varies greatly within the high-grade (HG) group. Examining the molecular characteristics of PMP instances might assist in more accurately classifying patients and anticipating how they will respond to therapy. Arjona Sanchez [30] analyzed the Ki-67 proliferation rate and P53 overexpression in tissue samples from our group of patients with high-grade pseudomyxoma peritonei. The study concluded that dividing the HG-PMP category of the PSOGI classification based on the Ki-67 proliferation index results in two distinct subcategories that show substantial variations in overall survival (OS) and disease-free survival (DFS), indicating a high prognostic value.

In a thorough analysis, Wang [31] looked into 830 instances of PMP to study the somatic mutational landscape and prognosis predictive variables of ovarian PMP. Among them, there were 16 patients found with PMP originating from the ovary. Whole-exome sequencing (WES) was conducted on 12 patients using formalin-fixed, paraffin-embedded (FFPE) tissue samples. Of the patients, 25% (3/12) had mutations in cancer driver genes such as TP53, ATM, and SETD2, while 16.7% (2/12) had mutations in cancer driver genes such as ATRX, EP300, FGFR2, KRAS, NOCR1, and RB1. The genes MUC16, BSN, PCNT, PPP2R5A, PRSS36, PTPRK, and SBF1 had the greatest mutation rates, ranging from 41.67% to 58.33%. The PI3K-Akt signaling pathway, human papillomavirus infection route, cell skeleton, cell adhesion, and extracellular matrix and membrane proteins were the primary pathways or functions impacted. The mutational profile of peritoneal mucinous carcinomatosis originating from the ovary was identified and showed specific characteristics different from those coming from the appendix. PTPRK, CA199, CCR, and PCI have the potential to forecast patient survival.

Dundr [32] concluded in a study on the mucinous tumors of the ovary that, although there was increasing understanding of the characteristics of ovarian mucinous tumors at both macroscopic and microscopic levels, along with a variety of immunohistochemical antibodies that could be helpful, there were still some tumors that could not be accurately classified, without close collaboration between their clinical and pathological aspects.

Tumor angiogenesis causes an increase in blood flow, resulting in elevated blood flow velocities (BFVels) and wall shear stress (WSS) in the arteries upstream. Pseudomyxoma peritonei (PMP) is a condition characterized by the growth of aberrant vascular networks atop existing ones in the peritoneum. Barral [33] determined that the amount of blood flow indicated the extent of the tumor in PMP, and that the wall shear stress in the superior mesenteric artery was an early and accurate vascular indicator of disease progression.

The genetic traits and mismatch repair (MMR) status of the original tumor and its related metastases in colorectal cancer (CRC) are often seen as closely aligned. Either the original or metastatic tumor may be used for assessing gene mutation and MMR status. Van der Meer [34] discovered that there was full agreement in biomarker expression between MMR-proficient colorectal cancer and their corresponding ovarian metastases. Testing of biomarkers in MMR-competent CRC tissue is enough, and further testing of metastatic ovarian tissue is unnecessary. Therapy response discrepancies between ovarian metastases and other metastases from CRC are unlikely due to genetic status differences.

Yang [35] believed that mass rupture and pseudomyxoma peritonei were significant risk factors for recurrence, but genetic alterations played a crucial role in promoting the growth. The case was that of a young woman with a borderline mucinous ovarian tumor who developed ipsilateral ovarian anaplastic carcinoma after 3 months, leading to metastases to the contralateral ovary and widespread dissemination in the pelvic and abdominal areas.

#### 3.2.2. Clinical and Pathology Findings

As shown by Yan [4], ovarian-derived PMP is very uncommon. To obtain an accurate diagnosis, it is necessary to examine consecutive sections of the appendix or suspicious tissue to rule out appendiceal mucinous neoplasms (see Figure 8 and Figure 9). This involves analyzing clinical aspects, surgical findings, histological characteristics, and particular biomarkers using immunohistochemistry. More so, as described by Fu [36], the histopathologic subtype and peritoneal cancer index (PCI) may be used as prognostic indicators in patients with mucinous ovarian carcinoma (MOC)-derived pseudomyxoma peritonei (PMP). Patients with advanced-stage illness (see Figure 10 and Figure 11) may benefit from achieving complete cytoreduction (CCR) 0/1.

Weitz [37] conducted a study in 2022 where culture and imaging models of ex vivo organotypic pseudomyxoma peritonei tumor slices from resected human tumor specimens were presented. The significance of this research is important due to the fact that studying PMP is difficult since it is rare, has few mouse models available, and has a histology characterized by mucinous and acellular features. This technology enables immediate visualization and examination of various tumor forms utilizing ex vivo organotypic slices taken from patients, maintaining the integrity of the tumor microenvironment (TME).

#### 3.2.3. Imaging Findings, Diagnosis, and Differential Diagnosis

Fonseca [38] emphasized that, despite its rarity, radiologists should be knowledgeable about the imaging characteristics, typical sites, and recurrence patterns of P. peritonei. It is important to discover the source of the main tumor. Further research is required to identify preoperative imaging markers that might predict surgical results and to describe the primary features of radiological recurrence. In his collection of 30 patients, the appendix was the most frequent location of the initial tumor (63.3%), followed by the ovaries (16.6%), with 16.6% of tumors having an unknown origin.

According to a study by Campos [39], PMP often has a mass impact on the mesentery and small bowel rather than directly affecting these tissues. Moreover, PMP may appear on ultrasound as intraperitoneal fluid that is very reflective, including echoes that remain stationary despite body movements, and showing many septations in a layered concentric pattern. The septations delineate the borders of the mucinous nodules. PMP often has a significant effect on the mesentery and small bowel rather than directly targeting these tissues. PMP may be visualized on ultrasonography as intraperitoneal fluid with high reflectivity, including echoes that are immobile despite body motion, and displaying many septations arranged in a layered concentric fashion. The septations define the boundaries of the mucinous nodules.

Raposo Rodriguez [40] tried to delineate the imaging characteristics of mucinous tumors in the abdomen and pelvis, which exhibited a uniform look upon imaging, regardless of their location within the body. The tumors resembled water upon ultrasonography, computed tomography, and magnetic resonance imaging, due to their high mucus content. Calcifications were often seen in mucin-producing tumors.

Noghabaei’s work [41] highlighted the constraints of sonography by detailing a unique instance of appendiceal mucocele in a young woman that resembled a pelvic tumor. The sonography misconstrued the tumor as a solid mass located near the uterus. It is crucial to diagnose appendiceal mucocele before surgery because of the potential for simultaneous gastrointestinal and ovarian malignancies. Qiao [42] demonstrated the significant practical value of CEUS in distinguishing between benign and malignant ovarian tumors. Additionally, CEUS might be beneficial for distinguishing between various conditions in the first phases of this illness.

In a study designed by Tanaka [43], OTAMT (ovarian teratoma-associated mucinous tumors) and ovarian metastases of AMT (appendiceal mucinous tumors (AMT)) both presented as multilocular cystic tumors with a homogeneous signal. A unilateral condition with intratumoral fat and a reduced appendix size may indicate the presence of OTAMT. According to a study by Ayadi [18], ultrasonography and CT scans are recommended for thorough examination, with abdominal MRI being a helpful technique for finding uncertain appendix abnormalities; these investigations should help in the orientation of a differential diagnosis between an appendiceal mucocele with PMP and a malignant ovarian tumor with peritoneal carcinomatosis. As appendiceal mucinous neoplasms with PMP are uncommon and may resemble ovarian tumors, Kostov [44] found that patients presenting with right pelvic masses should be evaluated for appendiceal tumors as part of the differential diagnosis. The clinical exam should, of course, be followed by CT scan, abdominal/vaginal ultrasound, and tumor marker levels (CEA, CA 19.9, CA-125) that help confirm the diagnosis. According to some study results [45,46], the CA-125/CEA ratio is a good way to distinguish between ovarian cancer and non-ovarian malignancies and, furthermore, in the case of a low Ca 125 and a high CEA endoscopy, assessment of the bowel would be indicated.

As the importance of a CT scan in the proper evaluation of this disease is emphasized various times in this article, we find it of uttermost importance to describe the common features of pseudomyxoma peritonei, according to a CT-scan description [47]. Likewise, the classic presentation of pseudomyxoma peritonei is as follows: loculated collections of fluid that build up along peritoneal surfaces, giving coated abdominal organs a scalloped look and omental caking. One may see an appendiceal mucocele, which is a mucinous appendiceal tumor. Due to the fact that appendiceal tumors are the most common cause, early peritoneal disease may only affect the lower right quadrant of the abdomen. Deposits tend to localize to the site of physiologic lymphatic absorption of ascites (omentum, underneath the right hemidiaphragm), and dependent areas (e.g., paracolic gutters, right retrohepatic space, lower pelvis). Before eventually causing bowel obstruction later in the disease, deposits usually spare the more mobile small bowel.

Peritoneal spread of ascending colon carcinoma has been evaluated. Namba [48] describes an uncommon instance of widespread spread of T2 colorectal carcinoma in the peritoneum without involvement of lymph nodes.

Brimo Alsaman [49] considered that the diagnosis of PMP might be difficult because of the variability of symptoms and signs across patients; many instances were asymptomatic and often found inadvertently during laparoscopy.

Another study that emphasized the difficulty of the differential between an appendiceal mucinous neoplasm that mimicked an ovarian tumor was the one conducted by Zhang [50], who found that frozen sections of both ovaries and of the appendix regardless of its appearance might improve the diagnosis. Raje [51] describes the case of a post-menopausal woman with endometrioid endometrial cancer who also had a low-grade appendiceal mucinous tumor with PMP.

Other types of lesions that can cause a doubt in diagnosis are the cystic formations in the renal pelvis, which can be benign, premalignant, or malignant and with a primary or secondary cause, as shown by Tamsin [52].

The variation in the types of lesions that might mimic a mucinous ovarian neoplasm with pseudomyxoma peritonei is broad, and it also includes a benign multicystic peritoneal mesothelioma, as explained by Altintas Taslicay [53]. Another clinical and pathological (differential) diagnosis is the one made with a ruptured primary mucinous cystadenoma of the spleen leading to mucinous ascites, as explained by Gupta [54].

Among the rarest elements to search for in the pursuit of a differential diagnosis (Yoshizaki [55]) is pseudomyxoma peritonei (PMP) originating from an intraductal papillary mucinous neoplasm (IPMN) of the pancreas. Prabhu’s case report [56] demonstrates a rare disease where peritoneal metastasis from gall bladder cancers usually has a poor prognosis. However, peritoneal dissemination from a low-grade mucinous neoplasm of the gall bladder (PMP) has a notably better prognosis because of the improved disease biology and treatment options.

When encountering a bilateral mucinous borderline ovarian tumor together with peritoneal implants or pseudomyxoma peritonei, it is advisable to search for a primary tumor originating from the digestive or pancreato-biliary tract by performing suitable investigations, according to Eymerit-Morin [57]. Intraoperatively, it is advisable to thoroughly sample suspected borderline ovarian tumors, focusing on solid regions and areas with vegetations.

Research designed by Aso retrospectively analyzed individuals who were sequentially recruited and diagnosed with PMP with an appendiceal origin. They had preoperative 18F-FDG PET/CT. The SUVmax was determined as the greatest SUVmax value in the abdomen, excluding the main site. SUVmax was compared with the pathological grade (low or high grade) of PMP tumors based on the World Health Organisation categorization. It was then analyzed in terms of the predicted cutoff point, sensitivity, specificity, and receiver operating characteristic. The research indicated that the maximum standardized uptake value (SUVmax) from preoperative 18F-FDG PET/CT scans correlated with the pathological grade in individuals diagnosed with PMP. Aspects on PET-CT scans of PMP can be found in Figure 12, Figure 13, Figure 14, Figure 15 and Figure 16.

Hegg [58] demonstrated the need to thoroughly examine surgical tissues for mucin, neoplastic cells on serosa, and microscopic perforation, since these factors have prognostic value.

In order to conduct a preoperative diagnosis, we consider that a patient with PMP would benefit from the following: a contrast-enhanced CT scan of the abdomen and the pelvis; this is because apart from reporting the radiological findings, it can also predict the likelihood of complete cytoreduction. Moreover, tumor markers, such as CEA, CA125, and Ca 19-9 can orient the clinician regarding the origin of the PMP. Furthermore, to increase the precision, if the operation is not imminent, as can occur in the case of a surgical emergency, then of course the panel of investigations can be completed using the following: abdominal ultrasound, colonoscopy, biopsy, blood tests, MRI scan, or even PET-CT.

This Section 3.2.3. can be seen summarized in Figure 17 (decision aid in the approach to a suspected case of pseudomyxoma peritonei).

#### 3.2.4. Treatment

Among the first treatment options that one might find as a solution for pseudomyxoma peritonei are HIPEC and cytoreductive surgery (CRS, using techniques such as Sugarbaker’s), as reinforced also by Manzanedo [59] and Narasimhan [60]. An equivalent term for cytoreductive surgery is “debulking”, as noticed during the literature search on the subject of pseudomyxoma peritonei. Bartoska [61] described a lethality of 0–12% and a morbidity of 27–56% for these two techniques and therefore considers HIPEC and CRS to be the gold standard in the treatment of *P. peritonei*. It is also Bartoska [62] who proposed a triple treatment for the same pathology, consisting of HIPEC, CRS, and EPIC (early postop intraperitoneal chemotherapy). The many assets emphasized in the literature on the subject of HIPEC and CRS also include the fact that it can be used in the case of recurrence. Regarding this latter subject (that of recurrence), Mercier [63] showed that preoperative chemotherapy and high-grade pathology were significant prognostic factors. More so, Conley [64] discovered that secondary cytoreductive surgery (CRS) plus hyperthermic intraperitoneal chemotherapy (HIPEC) might be a beneficial treatment choice for patients with recurrent primary mucinous ovarian cancer. Furthermore, Antos [65] found that second-look surgeries and preventive hyperthermic intraperitoneal chemotherapy (HIPEC) may positively impact the outcome after first complete tumor removal. Regarding the economic possibilities, Deo’s findings [66] suggest that it is possible to create an effective CRS and HIPEC program for PSM at government-funded hospitals in low middle-income countries with little resources. On the contrary, Souadka [67] looked into the challenging process generated by the implementation of such a program.

Laparoscopic cytoreductive surgery and hyperthermic intraperitoneal chemotherapy (L-CRS + HIPEC) have shown reduced duration of hospital stays and post-operative complications in small groups of carefully chosen patients. Analyzing a series of data studied by Arjona-Sanchez [68] the results showed that L-CRS + HIPEC is a safe and practical surgery for carefully chosen patients with modest peritoneal illness when conducted at specialized facilities. Psomiadou [69] also supported the use of minimally invasive surgery with HIPEC.

In parallel with the many advantages offered by HIPEC, there were also studies looking at the downsides of it. For instance, Kitai [70] emphasized that there was currently no clear standard procedure for HIPEC. The criteria used for selecting patients must be clearly defined in order to realize tangible advantages. Moreover, Robella [71] demonstrated that the perioperative risks of morbidity and death after CRS and HIPEC are similar to those of other significant gastrointestinal surgeries. CRS and HIPEC should be considered for carefully chosen patients (similar to Kitai’s opinions) seeking curative or life-extending therapy and should be conducted at specialized facilities with substantial case volumes. Naved [72] presented a dangerous side effect of HIPEC: following 48 h of CRS-HIPEC therapy, a patient of his exhibited hemodynamic instability and severe chest discomfort. An electrocardiogram revealed wide complex tachycardia with ST depression in leads V3–6. Severe systolic dysfunction with an Ejection Fraction (EF) of 20% and severe pulmonary hypertension were seen upon echocardiography. The diagnosis of Stress-induced Cardiomyopathy was confirmed by the use of the InterTAK Diagnostic Score. Patients with CRS-HIPEC have shown Stress-induced Cardiomyopathy, while no precise correlation between the two has been confirmed. This case report examined Stress-induced Cardiomyopathy as a complication of CRS-HIPEC. Another drawback of the CRS-HIPEC procedure is the presence of a notable number of early infective problems in patients undergoing CRS-HIPEC. Certain high-risk categories, such as those who have had small bowel resection and those with a higher number of resected organs, may need more intensive monitoring, as emphasized by Smibert [73]. During Pintado’s research [74], 77.1% of patients who had CRS and HIPEC experienced hematological issues in the postoperative phase. These difficulties were mostly mild and resolved on their own, without impacting death rates or length of hospital stay. Furthermore, Rubio-Lopez [75] discovered that the patient’s age, post-HIPEC potassium level, and pre-HIPEC glycemia are characteristics that might predict morbidity.

Regarding the safety of these two treatments (HIPEC + CRS), studies were conducted to show their utility, even in pregnancy, as explained by Papageorgiou [76], who concluded that pregnancy could be achieved in certain individuals after CRS and HIPEC. He also emphasized the importance of discussing assisted reproduction methods with the patient before the operation, ensuring that it does not impact overall survival or increase the chance of locoregional recurrence. Basso [77] provided a literature review on PMP discovered during pregnancy and discussed its therapy. Another aspect of his research focused on the treatment of diagnosing an ovarian tumor during pregnancy.

Patients with cancer who need cytoreductive surgery and hyperthermic intraperitoneal treatment may also have severe obesity. Some patients may be eligible for receiving bariatric surgery, such as sleeve gastrectomy, along with their cancer removal operation to address both conditions simultaneously. Two examples were documented by Aburahmah [78], where sleeve gastrectomy was performed concurrently with the main oncologic surgery in a single operation. The patients were monitored for an extended period, and their overall results were positive. They attained remission and satisfactory levels of weight reduction over their multiple years of follow-up consultations.

Hishida [79] looked into this subject and found that cytoreductive surgery remained beneficial for slow-growing and borderline malignant tumors including pseudomyxoma peritonei. In the same study, recent insights into tumor heterogeneity and clonal development suggested that some patients could achieve extended life via the combined impact of cytoreductive surgery and innovative systemic treatment. Brennan [80] addressed the importance of the correct use of each term and found that it was time to eliminate the word “debulking” from the medical literature on ovarian cancer and replace it with “cytoreductive surgery”. The same author emphasized that the phrase “optimal debulking” has evolved from referring to residual disease of less than 2 cm to today indicating “complete gross resection”, also known as “no macroscopic disease” or “zero-residual”. Although standardized descriptive terminology is lacking, the results of full surgical excision are evident. Bristow and colleagues conducted a meta-analysis of 81 papers, which revealed a 5.5% rise in median survival for every 10% increase in the percentage of women who had total tumor excision.

The peritoneal cavity is susceptible to containing peritoneal metastases, and it might help the clinical evolution if some anatomical formations were regularly removed. Nors [81] conducted research to see how often disease occurred in regularly resected CRS specimens. The results were that a significant number of standard surgical removals showed histologically confirmed peritoneal metastasis. Those findings support the regular removal of the umbilicus, ligamentum teres hepatis, ovaries, and larger omentum (see Figure 18 and Figure 19).

Total parietal peritonectomy (TPP) is a safe surgical method in the hands of a surgical expert in pseudomyxoma, and it can be used to treat peritoneal surface cancers and their recurrences. It has a low risk of grade IIIB morbidity and no treatment-related deaths, allowing for optimum operation, as presented by Saadeh [82]. From the same author, the investigation did not find any unusual recurrence sites, such as involvement of the abdominal muscles.

Hiraide [83] introduced the first case series of mFOLFOX6 in individuals with unresectable PMP and the first case series of systemic chemotherapy for Asian patients with unresectable PMP. Yazawa [84] discussed the utility of the multidisciplinary treatment.

Pressurized intraperitoneal aerosol chemotherapy (PIPAC) is a novel medication administration technique used in individuals with primary or secondary peritoneal carcinoma (PC). Using oxaliplatin intraperitoneally raises concerns about toxicity, particularly abdominal discomfort. Sgarbura [85] conducted research to evaluate the tolerance of PIPAC with oxaliplatin (PIPAC-Ox) in a significant number of patients and to determine the parameters associated with severe side effects, therapy termination, and reduced survival. The conclusion was that Oxaliplatin-based PIPAC seemed to be a secure therapy that provides effective symptom management and encouraging survival outcomes for individuals with advanced peritoneal illness. Ceribelli [86] considered PIPAC to be a safe and viable therapy for patients with peritoneal carcinomatosis who were originally not suitable for surgery, either to decrease tumor invasion or for palliative purposes to alleviate symptoms. Contraindications included intestinal blockage and several intraabdominal adhesions.

Rana [87] presented a case of multiple small bowel perforations caused by tumor lysis syndrome, which occurred as a complication of pseudomyxoma peritonei in a patient receiving intraperitoneal chemotherapy. Guchelaar [88] found that additional Phase II and Phase III studies were needed to determine the tolerability, survival rates, and impact on the quality of life of intraperitoneal chemotherapy as a treatment option for unresectable peritoneal disease.

Numerous variations of intraoperative chemotherapy have arisen, leading to ambiguity in selecting the best appropriate option. Mageed [89] wanted to increase awareness and take prompt actions to address the variability in managing peritoneal metastases via their review.

Huang [90] suggested Apatinib as a potential therapy for recurrent PMP after surgery; however, more validation is needed to corroborate his findings.

Performing percutaneous ultrasound-guided day-case aspiration of mucin for advanced and recurrent PMP utilizing a wide-bore drain is a safe and successful treatment. Borg [91] suggested that it may be used in palliative care or as a temporary measure before surgery for very symptomatic patients or when surgery is not feasible due to a reversible contraindication.

Sullivan [92] described superinfected PMP as an uncommon occurrence with little known instances. Implementing a tiered approach, including eliminating the peritoneal infection, with or without excision of the underlying tumor, followed by rehabilitation and final surgery seems to be a secure and efficient therapeutic method.

Lopez Basave [93] found that PCI and histopathology are prognostic factors for overall survival. Patients with ovarian tumors and a Peritoneal Cancer Index (PCI) of less than 15 have a greater overall survival (OS), comparable to those with pseudomyxomas. Patients with PCI < 15 have a greater RFS.

Emphasizing that CRS + HIPEC must be provided in tertiary centers (with a focus on pseudomyxoma) is crucial. If a surgeon who is not a PMP specialist notices PMP during surgery, they should take biopsies, drain ascites, and close. Following that, the patient needs to be referred to a PMP center for CRS + HIPEC.

## 4. Discussion

A precise and complete diagnosis is crucial as it reveals the cause and guides the selection of the most suitable treatment plan. It is important to note that, even in the absence of histological evidence of cancer tissue at the level of the appendix, we should still consider colorectal cancer as the main source. Intestinal-type ovarian mucinous tumors are recognized as an additional cause of PMP. To obtain an accurate diagnosis, it is necessary to examine many sections of the appendix or suspicious tissue to rule out appendiceal mucinous neoplasms. This involves analyzing clinical aspects, surgical findings, histological characteristics, and particular biomarkers using immunohistochemistry.

Ovarian teratoma-associated mucinous tumors (OTAMTs) have histological similarities to appendiceal or intestinal mucinous neoplasms rather than intrinsic ovarian mucinous neoplasms. Thus, OTAMTs may be incorrectly identified as metastatic ovarian tumors originating from primary appendiceal mucinous tumors (AMTs), notwithstanding histological identification. Treatment for a primary acute mesenteric thrombosis often involves appendectomy, ileocecal resection, or right hemicolectomy. An initial borderline or malignant ovarian tumor is often treated with complete hysterectomy, bilateral salpingo-oophorectomy, omentectomy, and optimum cytoreductive surgery of the intraperitoneal dissemination. Pre-operative imaging diagnosis may help differentiate between these two diseases and influence the choice of therapy. Recent investigations have provided a comprehensive understanding of the imaging characteristics of primary mucinous ovarian neoplasms and metastatic ovarian tumors originating from the intestines.

There is presently no agreement on the best therapy for progressive PMP. Nevertheless, HIPEC and CRS are considered the gold standard in tertiary centers trained and specialized in the treatment of pseudomyxoma peritonei. This association of therapies is deemed the appropriate approach, despite its potential accompanying morbidity. Regardless of the existence of other methods, the majority of practitioners have embraced this aggressive technique of completely removing all intra-abdominal and pelvic illness, followed by intraperitoneal “hyperthermic” chemotherapy using mitomycin and 5-FU.

## 5. Conclusions

Diagnosing PMP with an ovarian cause early and accurately is challenging because of its rarity.

Patients with PMP identified by imaging should have appendectomy, abdominal puncture, or laparotomy, and examination of both ovaries as standard procedure.

Scalloped glands, subepithelial clefts, cellular stroma, and histiocyte aggregates, along with immunohistochemical markers such as CK7, CK20, CDX2, and SATB2, can aid in determining the source of PMP and assessing its biological behavior from various origins (excluding teratoma-associated tumors).

Consensus has been achieved on several elements including pathological categorization, nomenclature, preoperative assessment, eligibility for surgery, maximum tumor removal, technical specifics of cytoreductive surgery, and classification system for severe adverse events. Controversies persist about the HIPEC regimen, systemic chemotherapy, and early postoperative intraperitoneal chemotherapy.

## Figures and Tables

**Figure 1 cancers-16-01446-f001:**
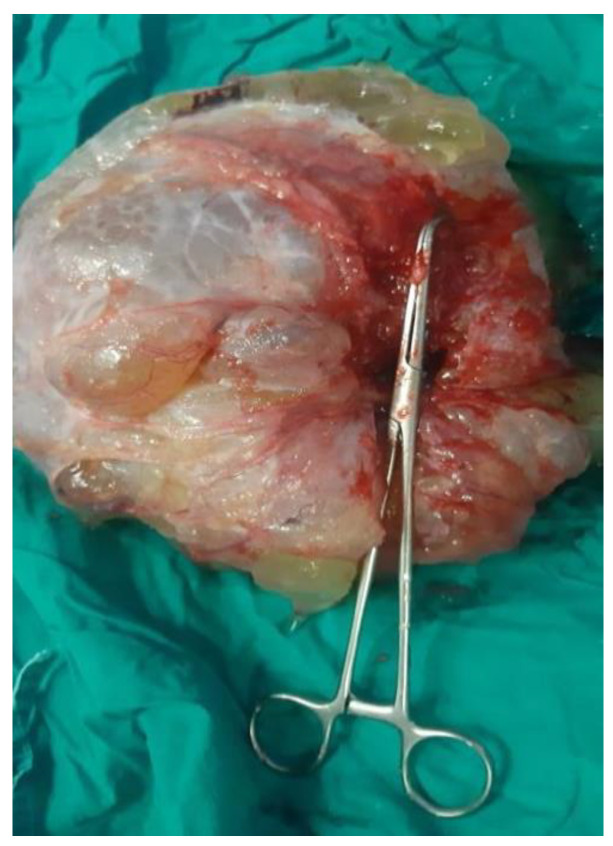
Intraoperative aspects of pseudomyxoma peritonei.

**Figure 2 cancers-16-01446-f002:**
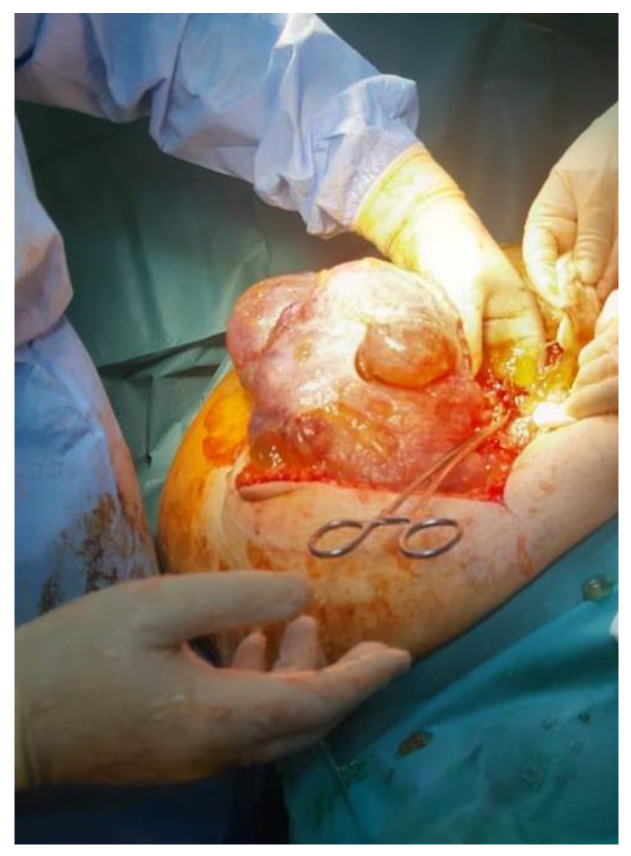
Intraoperative aspects of pseudomyxoma peritonei.

**Figure 3 cancers-16-01446-f003:**
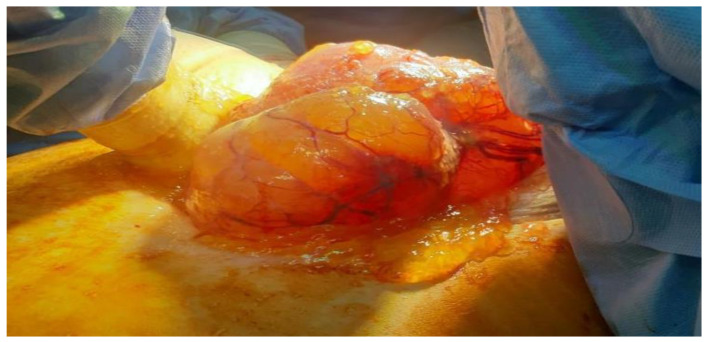
Intraoperative aspects of pseudomyxoma peritonei.

**Figure 4 cancers-16-01446-f004:**
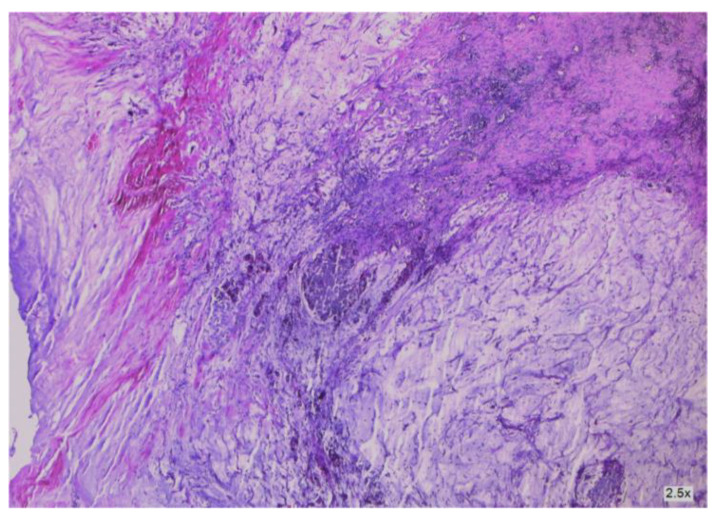
Areas of mucus in the wall of the appendix.

**Figure 5 cancers-16-01446-f005:**
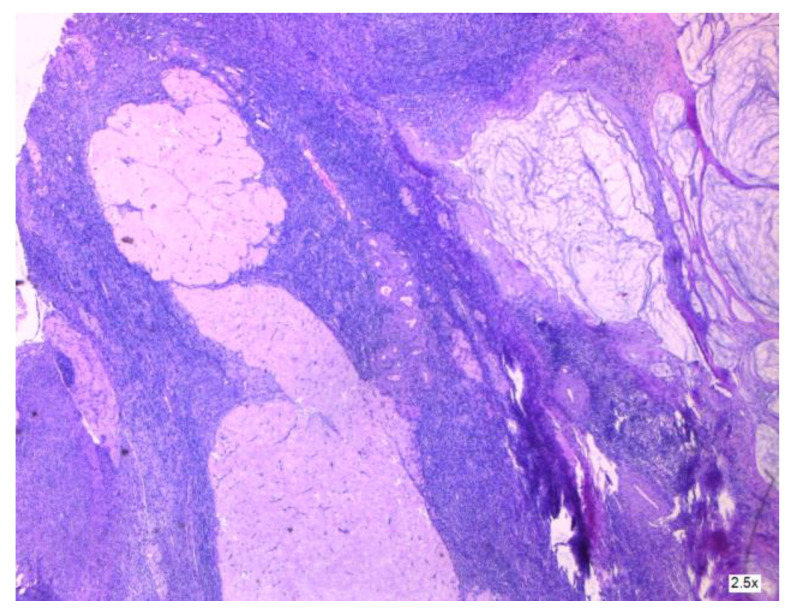
Right ovary with cancer spread in PMP.

**Figure 6 cancers-16-01446-f006:**
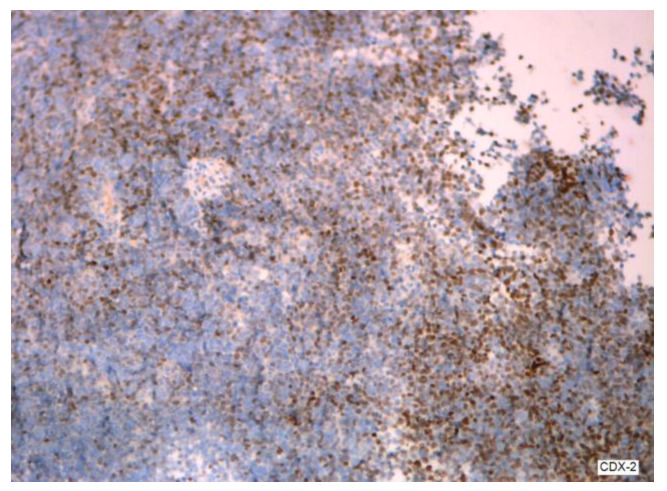
CDX2 in a PMP tumor.

**Figure 7 cancers-16-01446-f007:**
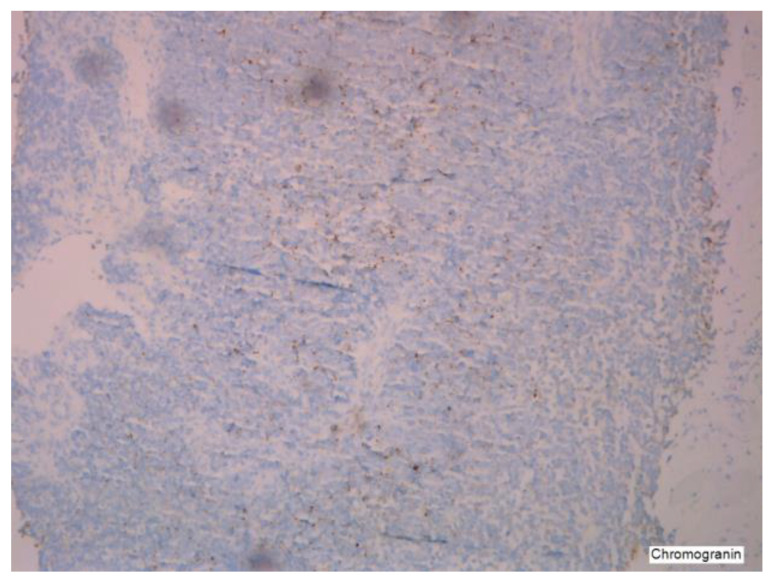
Chromogranin in a PMP tumor.

**Figure 8 cancers-16-01446-f008:**
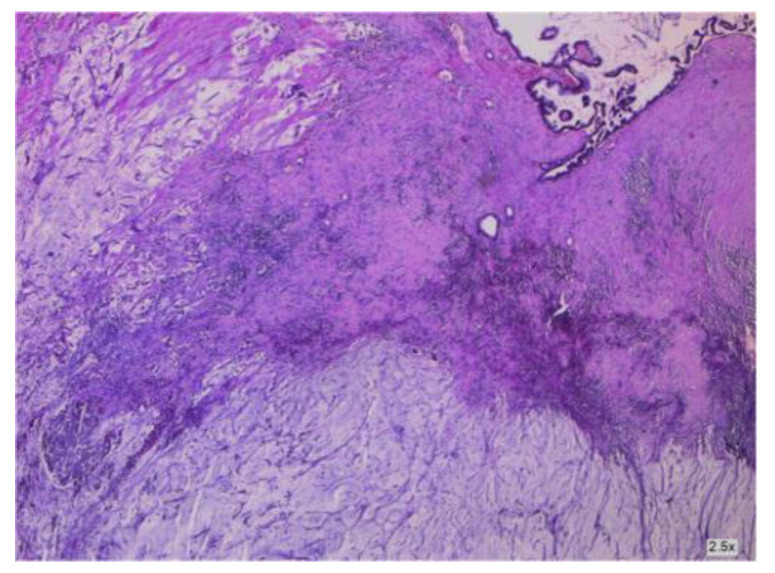
Cancer-infiltrated appendiceal wall.

**Figure 9 cancers-16-01446-f009:**
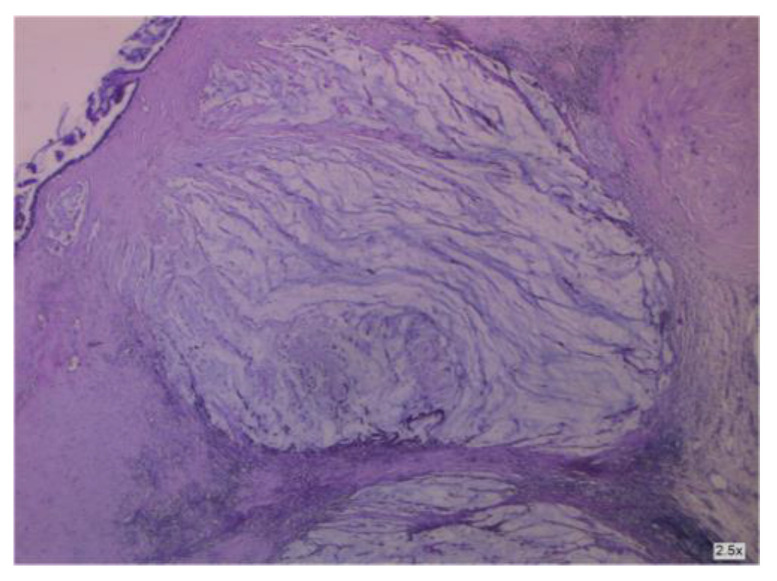
Cancer-infiltrated appendiceal wall.

**Figure 10 cancers-16-01446-f010:**
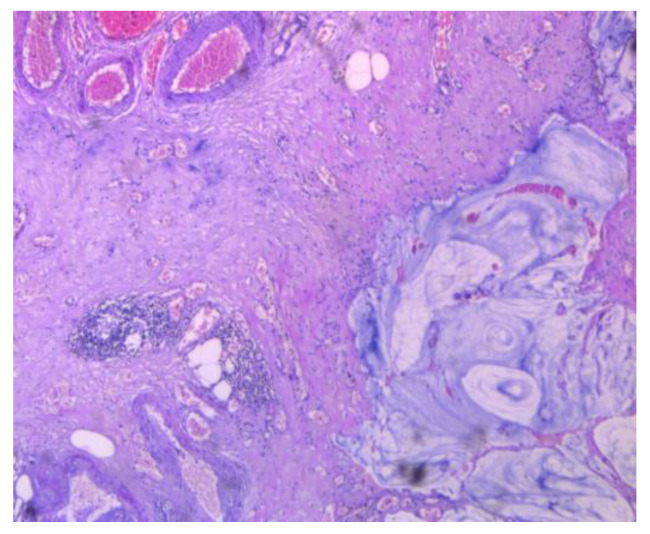
Tumor infiltrating the broad ligament, in PMP.

**Figure 11 cancers-16-01446-f011:**
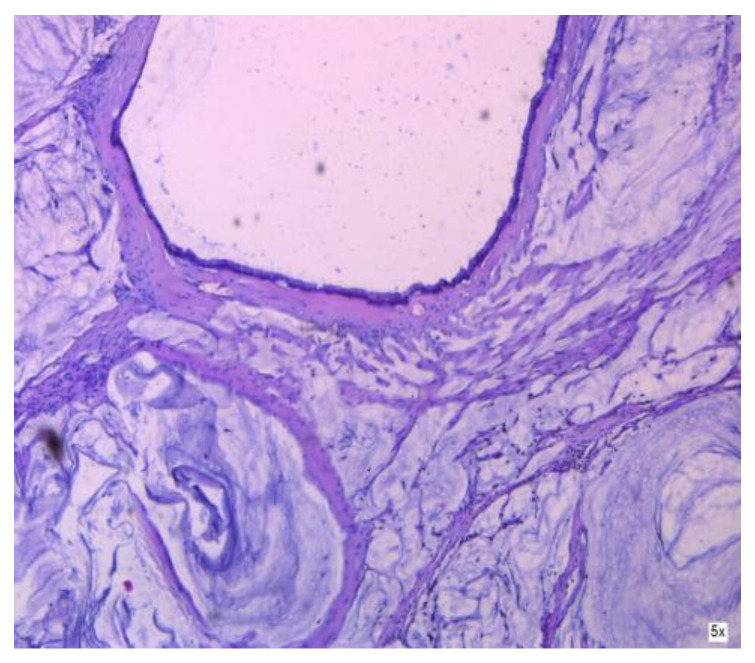
Tumor infiltrating the peritoneum, in PMP.

**Figure 12 cancers-16-01446-f012:**
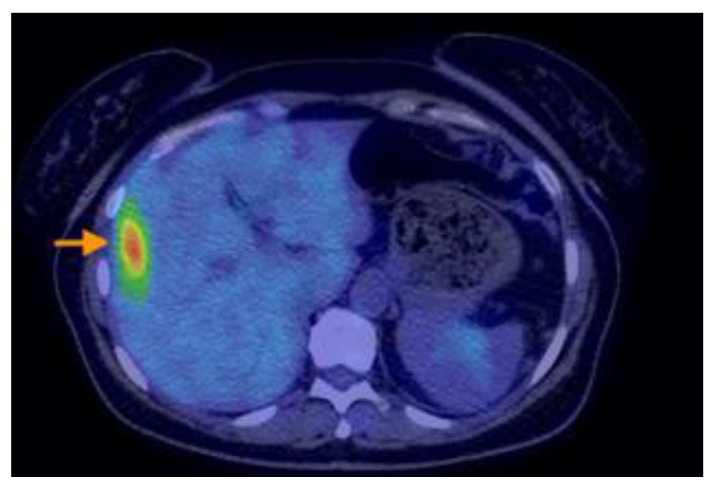
A 50-year-old woman with right ovarian mucinous carcinoma operated upon, who underwent chemotherapy, postoperatively. The PET-CT scan performed six months after completion of chemotherapy revealed multiple hypodense and cystic lesions on the visceral and diaphragmatic surfaces of the liver(see arrow), as well as throughout intraperitoneal spaces and mesentery, with moderate F18-FDG uptake.

**Figure 13 cancers-16-01446-f013:**
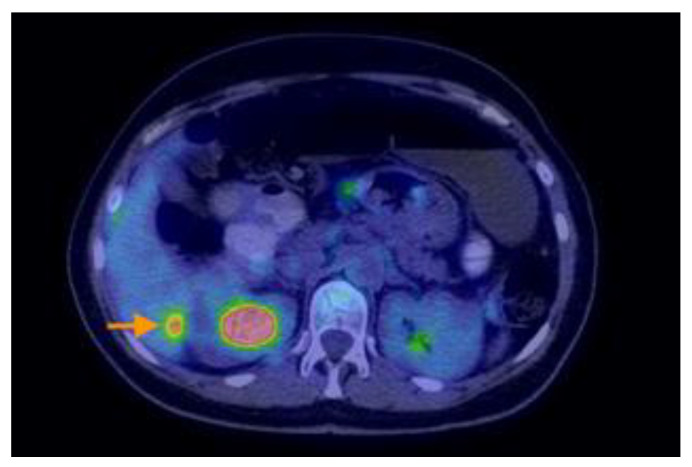
A 50-year-old woman with right ovarian mucinous carcinoma operated upon, who underwent chemotherapy, postoperatively. The PET-CT scan performed six months after completion of chemotherapy revealed multiple hypodense and cystic lesions on the visceral and diaphragmatic surfaces of the liver(see arrow), as well as throughout intraperitoneal spaces and mesentery, with moderate F18-FDG uptake.

**Figure 14 cancers-16-01446-f014:**
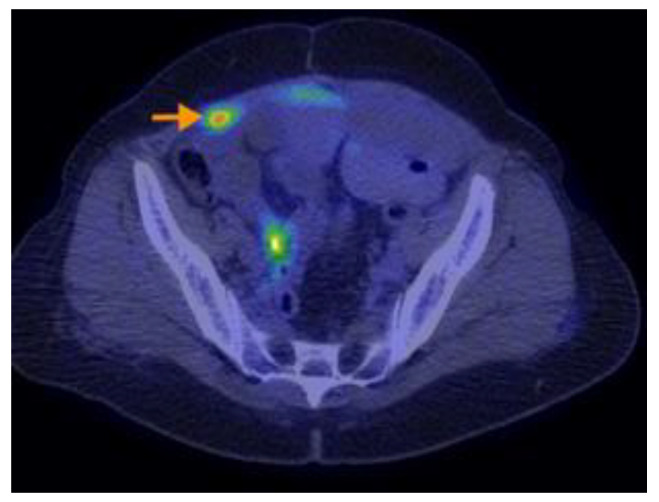
A 73-year-old woman with mucinous appendiceal carcinoma operated upon. PET-CT exam performed for suspicion of relapse, three years after completion of chemotherapy, shows multiple loculated collections of fluid accumulated along the peritoneal surfaces of the liver, intraperitoneal, and in the paracolic gutters and lower pelvis, presenting low or moderate F18-FDG uptake. Similar lesions with scattered calcifications are visible in the abdominal wall eventration.

**Figure 15 cancers-16-01446-f015:**
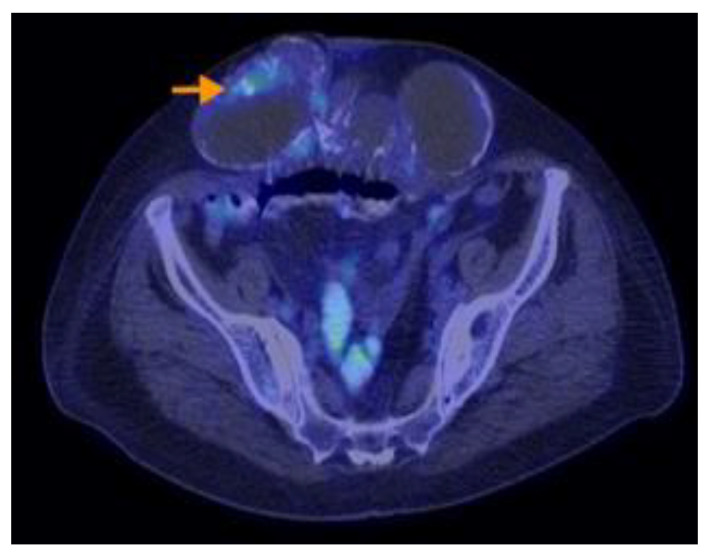
A 73-year-old woman with mucinous appendiceal carcinoma operated upon. PET-CT exam performed for suspicion of relapse, three years after completion of chemotherapy, shows multiple loculated collections of fluid accumulated along the peritoneal surfaces of the liver, intraperitoneal, and in the paracolic gutters and lower pelvis, presenting low or moderate F18-FDG uptake. Similar lesions with scattered calcifications are visible in the abdominal wall eventration.

**Figure 16 cancers-16-01446-f016:**
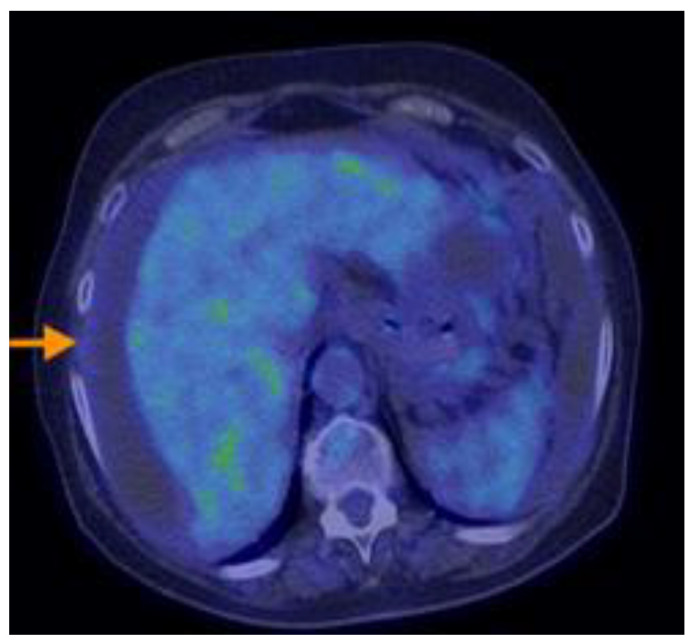
A 73-year-old woman with mucinous appendiceal carcinoma operated upon. PET-CT exam performed for suspicion of relapse, three years after completion of chemotherapy, shows multiple loculated collections of fluid accumulated along the peritoneal surfaces of the liver, intraperitoneal, and in the paracolic gutters and lower pelvis, presenting low or moderate F18-FDG uptake. Similar lesions with scattered calcifications are visible in the abdominal wall eventration.

**Figure 17 cancers-16-01446-f017:**
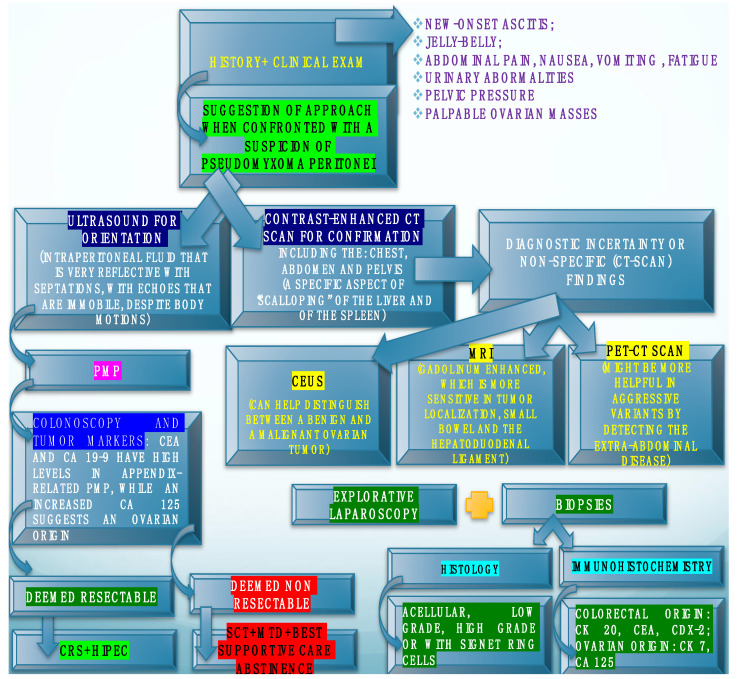
Decision aid in the approach to a suspected case of pseudomyxoma peritonei.

**Figure 18 cancers-16-01446-f018:**
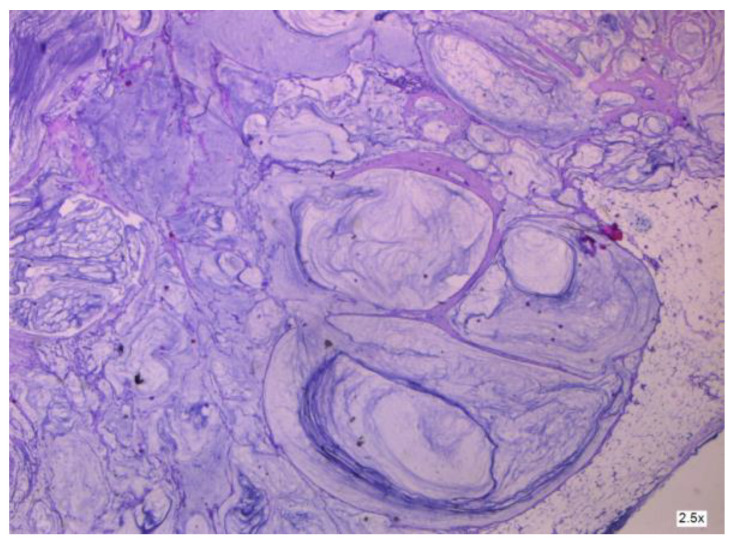
Tumor infiltration of the larger omentum.

**Figure 19 cancers-16-01446-f019:**
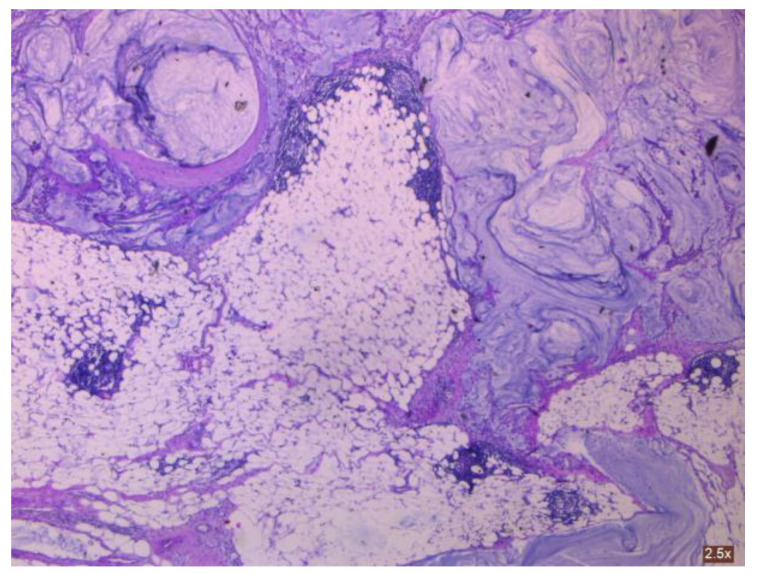
Tumor infiltration of the larger omentum.

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
