# Peer review of "Ovarian Causes of Pseudomyxoma Peritonei (PMP)—A Literature Review"

_cancers, 2024, doi:10.3390/cancers16081446_

Round 1
Reviewer 1 Report
Comments and Suggestions for Authors
Comments for authors
Section 3.1.2 Are the authors satisfied these case reports are ovarian origin? Were gastro-intestinal primary cancers excluded?
Line 116-117 The patient had a successful recovery and was released from the hospital on the seventh day after the procedure.
Suggest the word ‘discharged’ rather than released
Section 3.1.3 The authors should expand upon suggested protocols for intra-operative decision-making when faced with likely PMP from colon origin. Do they recommend peritoneal biopsies, appendicectomy and drainage ascites? Pseudomyxoma Centres would typically recommend that, if clinically disease appears to be metastatic colon PMP that surgery is performed with HIPEC in Pseudomyxoma Centre, with minimal disruption of tissue planes by index surgery.
Lines 158-160 How do the authors propose patients should be investigated to have a pre-operative diagnosis?
Lines 188-190 Mucinous borderline tumours infrequently recur, and when they do it tends to be late.
Sections 3.2.1 and 3.2.2 I would recommend review of these sections by a gynae histopathologist
Lines 322-324 Reference could be made to the importance of CA125:CEA ratio and the cut off value of 25 for likely of ovarian origin. Pre-operative of endoscopy to evaluate bowel when ratio low should be mentioned. Also, the authors should comment on the radiological features of pseudomyxoma, such as ‘scalloping’ of the liver.
Section 3.2.3 very wordy, may be helped with a decision-tree or flowchart
Section 3.2.4 The authors need to be clear that CRS+HIPEC needs to be delivered in tertiary centre specialising in pseudomyxoma. If PMP is incidentall noted at surgery, if the surgeon is not a PMP specialist, they should take biopsies, drain ascites and close. The patient should then be referred to a PMP centre for CRS + HIPEC.
Sentence in Line 398 can be removed
Line 442 debulking is also known as cytoreductive. Try and use same terminology throughout paper. Earlier authors used CRS
Lines 462 authors should clarify it is safe in hands of surgical expert in pseudomyxoma
Line 510 what does appendix residue mean?
Lines 528-533 CRS and HIPEC in expert tertiary surgical centre is gold standard. This should be made clear.
Comments on the Quality of English Languagethese are included in the comments to authors
Author Response
First of all, thank you for taking the time to review our manuscript! Secondly, we would like to mention that we greatly appreciate the opportunity we have been given to improve on our work, with the help of the indications and hints provided by the reviewers. We sincerely hope that we were able to successfully make the modifications asked and please know that we are trying to address each matter individually, in order not to miss anything, this is why we have thought that a table would pinpoint the modifications requested! You will find the table in the attached word document. Respectfully yours, the authors.

Reviewer 2 Report
Comments and Suggestions for Authors
Thanks to the authors for giving me the opportunity to review their manuscript. It is very well written, very complete. Pseudomyxoma peritonei (PMP) is a clinical entity defined by the intraperitoneal accumulation of mucinous ascites and mucinous tumor deposits originating from an intra-abdominal primary tumor. Almost all PMP are a consequence of an appendicular mucinous neoplasm; however, in 3% of PMP, the primary mucinous tumor is of ovarian origin. This remains difficult to prove, as ovaries are commonly invaded in appendiceal PMP. In ovarian PMP (OPMP), the tumor may be of germ cell origin (mucinous neoplasia arising from a mature teratomatous gastrointestinal-type epithelium) or of ovarian epithelial category (cystadenoma, borderline or malignant mucinous neoplasms).
I have a comment:
Accurate diagnosis of OPMP requires clinical, radiological and gastrointestinal endoscopic assessments as well as a histopathological examination of the appendix to exclude a primary digestive origin or an appendiceal mucinous neoplasia. I think that authors should specify in their manuscript the need to perform a gastro-colonoscopy to exclude any digestive lesions
Comments on the Quality of English Language/
Author Response
Respected Reviewer2, Thank you very much for taking the time to read our manuscript and give us your opinion! We have included in this revised version the importance of the gastro-colonoscopy to rule out gastrointestinal lesions, as you suggested. Respectfully yours, the Authors.
Round 2
Reviewer 1 Report
Comments and Suggestions for Authors
Authors have made a good attempt to address all the critiques and comments. Section 3.2.3 still very wordy. I cannot see the decision-aid.
Author Response
Respected Reviewer 1, I have added an illustration(Picture 17) which summarizes this section (3.2.3.), under the form of a decision-aid. Thank you for your time, patience and, of course, for your suggestions! Sincerely yours, dr Sinziana Ionescu